# Short-Term Adaptations in Skeletal Muscle Mitochondrial Oxidative Capacity and Metabolic Pathways to Breaking up Sedentary Behaviors in Overweight or Obese Adults

**DOI:** 10.3390/nu14030454

**Published:** 2022-01-20

**Authors:** Nathan P. De Jong, Michael C. Rudolph, Matthew R. Jackman, Rachel R. Sharp, Ken Jones, Julie Houck, Zhaoxing Pan, Jane E. B. Reusch, Paul S. MacLean, Daniel H. Bessesen, Audrey Bergouignan

**Affiliations:** 1Division of Endocrinology, Metabolism, Diabetes and Anschutz Health and Wellness Center, University of Colorado School of Medicine, Aurora, CO 80045, USA; nathan.dejong@cuanschutz.edu (N.P.D.J.); daniel.bessesen@cuanschutz.edu (D.H.B.); 2Department of Physiology, Harold Hamm Diabetes Center, Oklahoma University Health Sciences Center, Oklahoma City, OK 73104, USA; Michael-Rudolph@OUHSC.edu (M.C.R.); rsharp@unc.edu (R.R.S.); 3Division of Endocrinology, Metabolism, and Diabetes, Department of Medicine, Anschutz Medical Campus, University of Colorado, Aurora, CO 80045, USA; matthew.jackman@cuanschutz.edu (M.R.J.); julie.houck@cuanschutz.edu (J.H.); Paul.maclean@cuanschutz.edu (P.S.M.); 4Laboratory of Molecular Biology and Cytometric Research, Oklahoma University Health Sciences Center, Oklahoma City, OK 73104, USA; ken-jones@ouhsc.edu; 5Department of Biostatistics and Informatics, Anschutz Medical Campus, University of Colorado, Aurora, CO 80045, USA; Zhaoxing.Pan@CUANSCHUTZ.EDU; 6Center for Women’s Health Research, University of Colorado School of Medicine, Aurora, CO 80045, USA; Jane.reusch@cuanschutz.edu; 7Department of Medicine, University of Colorado, Anschutz Medical Campus, Aurora, CO 80045, USA; 8Rocky Mountain Regional Veterans Affairs Medical Center, Aurora, CO 80045, USA; 9Denver Health Medical Center, Denver, CO 80204, USA; 10Institut Pluridisciplinaire Hubert Curien, Université de Strasbourg, Centre National de la Recherche Scientifique (CNRS), UMR7178, 67037 Strasbourg, France

**Keywords:** sedentary behavior, mitochondria, skeletal muscle molecular adaptations, mRNA sequencing, active breaks, obesity

## Abstract

Breaking up sedentary behavior with short-frequent bouts of physical activity (PA) differentially influences metabolic health compared with the performance of a single-continuous bout of PA matched for total active time. However, the underlying mechanisms are unknown. We compared skeletal muscle mitochondrial respiration (high-resolution respirometry) and molecular adaptations (RNA sequencing) following 4-day exposure to breaks vs. energy-matched single-continuous PA bout in inactive adults with overweight/obesity. Participants (9M/10F, 32.2 ± 6.4 years, 30.3 ± 3.0 kg/m^2^) completed three 4-day interventions of a randomized cross-over study: SED, sedentary control; MICRO, 5 min brisk walking each hour for 9 h; ONE: 45 min/d continuous brisk walking bout. Fasted muscle biopsies were collected on day 5. Mitochondrial coupling in the presence of lipid-associated substrates was higher after ONE (4.8 ± 2.5) compared to MICRO (3.1 ± 1.1, *p* = 0.02) and SED (2.3 ± 1.0, *p* = 0.001). Respiratory rates did not differ across groups with carbohydrate-associated substrates. In pathways associated with muscle contraction transcription signaling, ONE and MICRO similarly enhanced Oxidative Phosphorylation and Sirtuin Signaling expression (*p* < 0.0001, for both). However, ONE (*p* < 0.001, for all), but not MICRO, had greater pathway enrichment, including Ca++, mTOR, AMPK, and HIF1α signaling, than SED. Although breaking up sedentary behavior triggered skeletal muscle molecular adaptations favoring oxidative capacity, it did not improve mitochondrial function over the short term.

## 1. Introduction

Prolonged sedentary behaviors (SB), defined as “any waking behavior characterized by an energy expenditure ≤ 1.5 metabolic equivalents while in a sitting, reclining, or lying posture” [1], are detrimental for health. It has been negatively associated with common risk factors for cardiometabolic disease, such as waist circumference, body mass index (BMI, kg/m^2^), 2 h postprandial glycemia, fasting plasma glucose, triglycerides (TG), and C-reactive protein. Conversely, frequently breaking up SB with short bouts of physical activity (PA) confers metabolic health benefits independent of total time spent in moderate to vigorous physical activity (MVPA) and total daily sitting time [2,3]. It is now well established that acute (≤12 h) and short-term (≤4 days) exposures to active breaks reduce postprandial glucose and insulin concentrations compared to a sitting control in adults who are lean, with obesity, or with type 2 diabetes [4,5,6,7,8,9,10,11,12].

To investigate the independent effects of the active breaks, the metabolic response of short-frequent PA bouts performed throughout the day have been compared to a time-matched single-continuous bout. Those studies have highlighted that for same total active time and/or energy expenditure, differential metabolic effects are elicited by these different PA modalities. Short-frequent bouts of PA decrease postprandial plasma glucose and insulin concentrations, while a single-continuous bout of PA reduces postprandial TG levels, both compared to prolonged sedentary periods [8,11]. In a randomized cross-over trial, we further showed that breaking up SB with short-frequent bouts of moderate intensity walking increases 24 h carbohydrate (CHO) oxidation, while a single-continuous bout of moderate intensity walking increases 24 h fatty acid (FA) oxidation in physically inactive adults with overweight or obesity [13]. However, the mechanisms underlying these differential metabolic effects are poorly known [14].

So far, only two human clinical trials have investigated some of the underlying mechanisms in skeletal muscle, which is a critical site for glucose disposal and nutrient metabolism both at rest and during PA [15]. Acute light- and moderate intensity walking breaks (2 min bouts every 20 min for 5 h) have been associated with differentially expressed genes involved in CHO and lipid metabolism (pyruvate dehydrogenase kinase 4 [PDK4] and nicotinamide N-methyltransferase [NNMT], respectively) as well as in cellular development, growth, and proliferation (myogenic factor 6 [MYF6], epithelial membrane protein 1 [EMP1], and dynein, light chain, LC8-type 1, transcript variant 1 [DYNLL1]) in adults with overweight or obesity [16]. Performing frequent active breaks for 3 days (2 min bouts every 20 min for 6 h/day) was shown to activate both non-oxidative and oxidative glucose uptake via insulin-independent and dependent pathways [17], which may optimize insulin action and both glucose oxidation and storage [18]. It remains unclear whether these molecular adaptations align with, or are different from, those triggered by a time-matched single-continuous bout of PA, and whether they can explain the differential metabolic responses observed at the whole-body level [19].

In this ancillary study, we tested the hypothesis that 4 days of breaking up SB with short-frequent bouts of PA would enhance mitochondrial respiration in the presence of CHO-associated substrates and skeletal muscle would increase expression profiles favoring CHO metabolism. Because prolonged PA is known to be associated with increases in whole-body fat oxidation rates, especially in post-absorptive conditions [20], we further hypothesized that after 4 days of a daily single-continuous bout of PA matched for total active time would result in greater lipid-linked mitochondrial respiration and expression of genes associated with the regulation of lipid metabolism. Our findings provide muscle-specific evidence that molecular and functional adaptations depend on the terms of PA, i.e., frequency, duration and/or volume, and could contribute to whole-body energetics to improve metabolic health outcomes.

## 2. Methods

### 2.1. Experimental Protocol

The detailed description of the study design and main outcomes have been published previously [13]. Following the screening visit, three interventions were completed by each participant in a randomized balanced order. Each intervention period lasted 4 days. The first 3 days were in free-living conditions and the 4th day was in a whole-room calorimeter. All interventions were separated by a 28-day washout period. Study procedures were performed during the same phase of the menstrual cycle in all women. All the visits were conducted at the Clinical and Translational Research Center of University of Colorado Hospital. This study was approved by the Colorado Multiple Institutional Review Board and is listed at ClinicalTrials.gov: NCT02258438 posted 10-07-2014.

### 2.2. Participants

Study volunteers were recruited from October 2014 to October 2016. Inclusion criteria included: age 19–45 years, BMI 27–33 kg/m^2^, weight stable for ≥3 months, and being sedentary defined as sitting ≥ 6 h/day. Women were premenopausal, but could use oral contraceptives. Exclusion criteria included clinically diagnosed diabetes, consumption of glucose- and/or lipid-lowering medication, dyslipidemia, smoking, or being physically active (>self-reported 150 min/wk MVPA).

### 2.3. Screening Visit

Written informed consent was obtained at the screening visit. Following verification of eligibility criteria, participants completed a treadmill walk test to determine a self-perceived moderate intensity walking pace. Participants were asked to walk at a similar moderate intensity pace in free-living conditions and completed the same treadmill walking pace in the whole-room calorimeter.

### 2.4. Study Interventions

Participants were randomized to one of three possible trial condition sequences using balanced blocks prepared separately for male and female participants. The study statistician (Z.P.) prepared the computer generated randomization assignments and placed them in sealed envelopes that were opened the day before the first intervention to reveal the order of study interventions.

Sedentary behaviors (SED): During the 3-day free-living period, participants were asked to refrain from structured exercise and maintain habitual levels of daily PA and SB. On day 4, participants remained sedentary in the whole-room calorimeter.

Sedentary + single-continuous bout of walking (ONE): During the 3-day free-living period, participants were instructed to perform 45 min of moderate intensity walking once per day and to maintain habitual levels of daily PA and SB the remainder of the day. On day 4, participants performed one 45 min bout of moderate intensity walking on a treadmill at 10:00 AM, but otherwise remained sedentary in the whole-room calorimeter.

Sedentary + microbouts of walking (MICRO): During the 3-day free-living period, participants were instructed to perform a 5 min bout of moderate intensity walking each hour for 9 consecutive hours and to maintain habitual levels of daily PA and SB the remainder of the day. On day 4, participants performed a 5 min bout of moderate intensity treadmill walking every hour for 9 consecutive hours from 10:00 AM to 6:00 PM and otherwise remained sedentary in the whole-room calorimeter.

Compliance was verified with objectively measured time spent sitting and lying, stepping, and daily steps with ActivPAL™ PA monitors (PAL Technologies Ltd., Glasgow, Scotland) and PA intensity was confirmed with ActiGraph GT3X tri-axial accelerometer (ActiGraph, Pensacola, FL, USA). Data on time spent sedentary and daily patterns of PA have been previously published [21].

### 2.5. Standardized Diet

Each participant consumed a standardized diet for 3 days leading up to the whole-room calorimeter study visit. The macronutrient content of the control diet was 30% fat, 55% CHO, and 15% protein of total energy intake. On day 4, the macronutrient composition was 20% fat, 65% CHO and 15% protein. Daily energy needs were calculated based on resting metabolic rate and fat-free mass measured by dual energy X-ray absorptiometry (DXA, Hologic Delphi-W, Bedford, MA, USA), as previously reported [13]. By design, participants received the same amount of food across the three interventions and as a result were in similar state of negative energy balance in the two PA trials [13].

### 2.6. Skeletal Muscle Biopsy

Following an overnight fast, a vastus lateralis skeletal muscle biopsy was collected under local anesthesia (2% wt/vol lidocaine HCl) using the Bergstrom biopsy needle technique on the morning of day 5. From the total biopsy sample, ~10–20 mg was placed immediately in ice-cold BIOPS solution (10 mM Ca–EGTA buffer, 0.1 M free calcium, 20 mM imidazole, 20 mM taurine, 50 mM potassium 2-[N-morpholino]-ethanesulfonic acid, 0.5 mM dithiothreitol, 6.56 mM MgCl2, 5.77 mM ATP, and 15 mM phosphocreatine [PCr], pH 7.1) and ~40–60 mg was snap frozen in liquid nitrogen and stored at −80 °C for future analysis.

### 2.7. Preparation of Permeabilized Muscle Fiber Bundles

Muscle fiber bundles were gently teased apart in a petri dish containing ice-cold BIOPS solution with fine-nosed forceps and a dissecting microscope. Approximately 2–5 mg fiber bundles were permeabilized in saponin (5 mg/mL) for 30 min at 4 °C and then washed once for 10 min at 4 °C with Mir05 respiration medium (0.5 mM EGTA, 3 mM MgCl_2_·6H_2_O, 60 mM K-lactobionate, 20 mM taurine, 10 mM KH2PO4, 20 mM HEPES, 110 mM sucrose, and 1 g/L BSA, pH 7.1), all on an orbital shaker [22]. The permeabilized muscle fiber bundles were blotted, weighed, and placed in the high-resolution respirometry (HRR) chambers of an Oxygraph 2K (Oroboros Inc., Innsbruck, Austria) for analysis.

### 2.8. Skeletal Muscle Mitochondrial Respiration Protocols

HRR of permeabilized muscle fibers offers an integrative ex vivo measure of the dynamics of coupled metabolic pathways [23]. Quantification of oxygen consumption in permeabilized fiber bundles was conducted at 37 °C in an oxygen concentration above 200 nmol O_2_/_mL_ over a ~2 h period. Substrate–uncoupler–inhibitor titration (SUIT) protocols for CHO and FA respiration was performed following stabilization of the O_2_ trace. The addition of cytochrome C was used to confirm the integrity of the outer mitochondrial membrane and the quality control threshold was set at ≤10% increase in respiration. Steady-state O_2_ flux for each respiratory state was determined and normalized to dry fiber bundle weight using Datlab 5 software (Oroboros Inc.).

In the CHO SUIT, LEAK state without adenylates respiration (i.e., substrate-driven non-phosphorylating inner membrane proton leak) was determined by the addition of saturating concentrations of pyruvate (2 M) and malate (0.4 M). State 3 coupled respiration supported by electron flux through complex I was measured with the addition of ADP (0.5 M). Oxidative phosphorylation (OXPHOS) was achieved by adding saturating concentrations of glutamate (2 M) and succinate (1 M). The addition of oligomycin (5 mM) inhibited ATP synthase and titrations of the uncoupler FCCP (carbonyl cyanide-p-trifluoromethoxyphenylhydrazone 0.1 mM) were performed to assess maximal electron transport system (ETS c1 + c2) respiration. Lastly, complex I inhibitor rotenone (1 mM) was added to measure the rate of respiration through complex II alone (ETS c2). To finish, malonic acid (2 M) and antimycin A (5 mM) were added to inhibit complex II and III, respectively. With total electron transport chain respiration inhibition, residual oxygen consumption was then quantified as a correction factor for all mitochondrial respiratory rates.

In the FA SUIT, LEAK state without adenylates respiration was measured by the addition of octanoylcarnitine (0.1 M) and malate (0.4 M). This was followed by ADP (0.5 mM) for State 3 coupled respiration and then pyruvate (2 M), glutamate (2 M) and succinate (1 M) for OXPHOS. The remainder of the FA SUIT protocol was the same as the CHO SUIT.

For both substrate-specific SUIT protocols, the respiratory control ratio (RCR) was calculated (State 3/LEAK) to determine the degree of coupled respiration and the flux control ratio (FCR; each respiratory state normalized to maximal respiratory rate, i.e., ETS c1 + c2) to determine oxygen flux in different respiratory control states.

### 2.9. RNA Sequencing

Samples were sequenced as previously described [24,25]. Briefly, frozen muscle samples were mechanically homogenized (~40–60 mg) in 1 mL of TRIzol, and RNA was isolated according to manufacturer’s instructions (Quick-DNA/RNA MiniPrep Plus Kit catalog# ZD7003, Research Products International, Mt. Prospect, IL, USA). RNA samples were submitted to the University of Colorado Denver’s Genomics and Microarray Core where quality was assessed with the Agilent 2100 Bioanalyzer both before and after library preparation, which used Nugen’s Universal Plus mRNA-seq kit according to manufacturer’s protocols (NuGEN, Redwood City, CA, USA). Sequencing was performed with the NovaSEQ6000 Ilumina sequencing platform as 150 nucleotide paired end reads, sequenced at 80 million paired reads per sample. Samples were run in two batches. Generated reads were trimmed, mapped to the human genome (GRCh38) using gSNAP; fragments per kilobase of transcript per million mapped reads (FPKM) were calculated using Cufflinks, and R was used for the discovery of differential gene expression [26,27]. Genes were removed if their FPKM expression was less than 5 in either group of the statistical comparisons or if the difference in means was >5. Raw mRNA sequencing data have been archived under the GEO accession number GSE186715.

### 2.10. Pathway Analysis

Canonical pathway and pathway comparison were conducted as previously described using IPA (Ingenuity Pathway Analysis, Qiagen, Hilden, Germany) [28]. Briefly, pathway analysis was performed using statistically significant differentially expressed genes (DEG) with *p*-values of <0.05. Comparison analyses of the DEG lists between PA modes (ONE v SED, MICRO v SED, and MICRO v ONE) were used to calculate biological pathways and functions that were in common among DEG lists. The output lists were ordered by pathway enrichment significance (Benjamini–Hochberg false discovery rate [FDR]) and pathways were colored by predicted activation Z-score (red = positive, blue = negative, white = N/A).

### 2.11. Data and Statistical Analysis

The appropriateness of normality assumption for analysis of variance (ANOVA) and linear mixed effects model analysis was verified using Shapiro–Wilk test and visual examination of residual plots. One-way ANOVA was used to test for differences in participant characteristics between three subgroups: participants from the parent study (N = 20) [13], participants included for Oroboros experiments to measure mitochondrial respiration in permeabilized muscle fibers (N = 19), and participants for RNAseq analysis for molecular analysis (N = 8). All initial respiratory states (LEAK, State 3, OXPHOS, ETS c1 + c2, and ETS c2) that qualified for inclusion in data analysis (≤10% change in respiration after the addition of cytochrome C) were assessed with the ROUT method to identify and remove outliers from the data set [29]. Linear mixed models tested for between-intervention differences in permeabilized muscle fiber respiratory states (LEAK, State 3, OXPHOS, ETS c1 + c2, and ETS c2), and ratios (RCR and FCR) with intervention as repeated effect, sequence, period, and intervention as fixed effects and participants as random effects with variance components as covariance structure. A *p* ≤ 0.1 was considered significant for model interactions. The least significant difference post hoc test was used to examine between-intervention differences; *p* ≤ 0.05 was considered significant. Differentially expressed genes were analyzed with unpaired *t*-test between (SED vs. MICRO, SED vs. ONE, and ONE vs. MICRO) followed by FDR, which was considered significant at ≤ 0.05 by the Benjamini and Hochberg multiple testing correction. Spearman correlational analysis stratified by intervention was performed on mitochondrial oxygen consumption-dependent variables with previously published whole-body metabolic outcomes as independent variables [13]. Linear mixed-effect model and between-group comparison of correlations were fit to the dependent variables for the three interventions simultaneously while accounting for the correlation of repeated observations from the same participants. *p* ≤ 0.1 was considered significant for model interactions and *p* ≤ 0.05 was considered significant for between-group differences in correlation. Analyses were performed using SAS software, version 9.4. (SAS Institute, North Carolina, USA) and in R. Data are provided as mean ± SD, unless specified otherwise.

## 3. Results

### 3.1. Participant Characteristics

As previously reported [13], 20 participants (*n* = 10 males and 10 females) completed all study-related procedures; their mean age was 32.4 ± 6.3 years and the BMI was 30.6 ± 2.9 kg/m^2^. A subsample of participants had fasting muscle biopsies harvested for measurement of mitochondrial oxygen consumption in permeabilized muscle fibers (N = 19, 9 males and 10 females) and a subsample for RNA sequencing and pathway analysis (N = 8, 4 males and 4 females). No differences in the participant characteristics were noted between the two subsamples and the sample from the parent clinical trial (Table 1).

### 3.2. Mitochondrial Capacity for Fat Oxidation Is Enhanced by a Single-Continuous Bout of PA, but Not by Multiple Short Bouts of PA

In the FA SUIT, no between-intervention differences in mitochondrial respiratory states were observed (Figure 1A), but RCR (i.e., degree of mitochondrial coupling) differed across interventions (intervention effect: *p* = 0.008). In the presence of FA-associated substrates (Oct + M + ADP; Figure 1E), the RCR was higher after completing 4 days of ONE (4.8 ± 2.5) compared to both SED (2.3 ± 1.0, *p* = 0.002) and MICRO (3.1 ± 1.1, *p* = 0.024). The higher coupling in the presence of FA-associated substrates was likely driven by a decrease in LEAK respiration. LEAK FCR (i.e., the contribution of mitochondrial respiratory state to overall flux; Figure 1C) differed across the three interventions (intervention effect: *p* = 0.020); it was lower in ONE (0.035 ± 0.008) compared to SED (0.075 ± 0.010, *p* = 0.006), but not to MICRO (0.052 ± 0.005, *p* = 0.21). In the CHO SUIT, there were no between-intervention differences observed for mitochondrial respiratory states (Figure 1B), FCR (Figure 1D), and RCR (Figure 1F).

### 3.3. Correlations between Skeletal Muscle Mitochondrial Oxidative Capacity and Whole-Body Metabolic Outcomes

Mitochondrial coupling (i.e., RCR) in the presence of lipid-associated substrates was positively associated with fasting insulin concentrations in both PA interventions (slope ONE = 0.65, *p* = 0.0008 and slope MICRO = 0.24, *p* = 0.02; Table 2). The slopes of these associations were significantly different from SED (slope ONE vs. SED: *p* = 0.015 and slope MICRO vs. SED: *p* = 0.05). Furthermore, ONE and MICRO differed (*p* = 0.03), the association was stronger in ONE than in MICRO. Similarly, mitochondrial coupling in the presence of lipid-associated substrates was positively associated with fasting insulin/glucose ratio, an index of insulin sensitivity, in both PA interventions (slope ONE = 58.11, *p* = 0.001 and MICRO = 23.24, *p* = 0.012). The slope for ONE was significantly different from SED (slope ONE vs. SED: *p* = 0.002) and the slope for MICRO tended to differ from SED (slope MICRO vs. SED: *p* = 0.07). Sleeping CHO oxidation (i.e., proxy of fasting CHO oxidation) was negatively correlated with CHO-supported LEAK respiration (slope = −36.06, *p* = 0.03) but not with OXPHOS and ETS c1 + c2 (*p* for interaction: *p* = 0.401 and *p* = 0.194, respectively). Sleeping fat oxidation (i.e., proxy of fasting fat oxidation) was not correlated with fat-supported LEAK, OXPHOS, and ETS c1 + c2 (*p* for interaction *p* = 0.919, *p* = 0.357, and *p* = 0.115, respectively).

### 3.4. Differential Gene Expression Leads to Similar Pathway Enrichment for MICRO and ONE with More Biological Pathways Enriched in ONE

Differentially expressed genes were calculated (FDR ≤ 0.05) relative to the SED condition, and 1126 DEG were identified in ONE and 678 DEG were identified in MICRO (Figure 2A). About 14% of DEG were shared between ONE and MICRO (Figure 2B). Ingenuity Pathway Analysis (IPA) software was used to evaluate biological pathways and functions (Figure 3), providing a global view of the gene expression differences for muscle adaptations to PA mode. Significantly different gene lists were analyzed using the Comparison Analysis tool in IPA, and the pathway enrichment scores (sum of −log10(FDR)) were calculated as an indication of the proportion of DEGs that populate a given pathway. The fold-changes in individual DEGs within a pathway were used to calculate whether a given pathway was predicted to be activated (red), inhibited (blue), or no prediction (white) [28]. This pathway analysis revealed similar DEG enrichment and predicted activation states for ONE and MICRO compared to SED. The Oxidative Phosphorylation pathway was predicted to be activated (SED vs. ONE: *p* < 0.0001; SED vs. MICRO: *p* < 0.0001) and the Sirtuin Signaling pathway was predicted to be inhibited (SED vs. ONE: *p* < 0.0001; SED vs. MICRO: *p* < 0.0001). Critical cellular energy and metabolic sensor pathways were significantly enriched following ONE (*p* < 0.01, parenthetical values) but not MICRO (*p* > 0.05) when compared to SED. For instance, the mTOR, ERK/MAPK, AMPK, Nrf2-mediated Oxidative Stress Response, Insulin Receptor, Calcium Signaling, Cecroptosis, and HIF1α pathways were enriched following ONE but not MICRO. Comparing both PA interventions (MICRO vs. ONE), several pathways were enriched, including Sirtuin Signaling, Oxidative Phosphorylation, mTOR, ERK/MAPK, AMPK, and NRF2-mediated Oxidative Stress Response pathways (*p* < 0.01 for all, parenthetical values). The Oxidative Phosphorylation pathway was predicted to be less activated following MICRO (MICRO vs. ONE activation z-score: −3.357, *p* < 0.001). Additionally, while these pathways were not enriched, Calcium Signaling and Necroptosis Signaling pathways were upregulated for MICRO (*p* < 0.05). All pathways expressed by the interventions are listed in Appendix A.

When comparing the significantly different genes modulated in the Oxidative Phosphorylation pathways for ONE or MICRO (relative to SED), more genes were similarly expressed in common between ONE and MICRO (N = 19/45), while some were unique only to MICRO (N = 10/45) or ONE (N = 16/45) (Figure 4A). Several genes regulated in common between both PA trials included downregulation of the mitochondria trafficking inesin-binding protein 1 (TRAK1) and apoptosis activator Bcl2-associated agonist of cell death (BCL2), mitochondrial ATP synthase subunit epsilon 5E (ATP5E), cytochrome c oxidase genes (COX6, -7A2 and -7B), and NADH dehydrogenase subcomplex subunits (NDUFA2, -4, -8, 11, and -13, NDUFB5, -9, NDUFS3, and NDUFV2), mitochondrial fission 1 protein (FIS1), and mitochondrial peroxiredoxin-5 (PDX5).

The Sirtuin Signaling pathway, critical for sensing cellular energy status and metabolism [30], was also regulated following both ONE and MICRO. Significantly different genes that were regulated in common between ONE and MICRO included forkhead box protein O4 (FOXO4, decreased), mitochondrial import inner membrane translocase subunits (TIMM8 increased, -23 increased and -50 increased, and PAM16 increased), and the mitochondrial cholesterol transport protein (TSPO increased) (Figure 4B). Significantly regulated genes in ONE included 5’-AMP-activated protein kinase catalytic subunit alpha-1 (PRKAA1, decreased), superoxide dismutase 1 and 2 (increased SOD1 and decreased SOD2), lipogenic transcription factor sterol regulatory element-binding protein 1 (SREBF1, decreased), and inflammatory response signal transducer and activator of transcription 3 (STAT3, decreased). Furthermore, ONE upregulated unique mitochondrial import inner membrane translocase subunits (TIMM13 and -17A), as well as mitochondrial import receptor subunits (TOMM6, -7, and 22) and several MAP kinase signal transduction genes (MAPK1, -3, and -12). In contrast to ONE, MICRO had unique regulation of mitochondrial isocitrate dehydrogenase 2 (IDH2, decreased), skeletal muscle autophagy regulators forkhead box protein O3 (FOXO3, decreased) and ubiquitin-like-conjugating enzyme (ATG3, increased), microtubule components (TUBA1A increased, TUBA1B increased, and TUBA8 increased), as well as complex V ATP synthase subunit gamma (ATP5C1 increased). All significant DEGs organized by mode of PA are presented in Appendix A.

## 4. Discussion

In this randomized cross-over study, we compared the functional and molecular adaptations in skeletal muscle biopsies to short-frequent bouts of PA spread throughout the day to a time-matched single-continuous bout of PA, both compared to sedentary control conditions, in adults with overweight or obesity. We identified changes in the regulation of gene expression and mitochondrial functional capacity that are in common to both PA interventions and others that are unique to either ONE or MICRO. This indicates that biological pathways in skeletal muscle differentially respond to the terms of PA, i.e., volume, frequency, and/or duration of the bouts.

Similar enrichment and activation of two key biological pathways known to regulate substrate oxidation, the Oxidative Phosphorylation pathway (including electron transport chain components) and the Sirtuin Signaling pathway, were observed after ONE and MICRO, suggesting that adaptations within these pathways depend on total daily active time rather than bout frequency and/or duration. These changes were observed 14 h after the last microbout of PA and 21 h after the single-continuous bout of PA, indicating a sustained response beyond the acute period. A potential regulator of muscle adaptation to acute forms of PA is FOXO4, which was downregulated following both ONE and MICRO. FOXO4 and family members are transcription factors known to carry out the inhibitory effects of insulin and IGF-1 [31], potentially supporting improved insulin sensitivity in skeletal muscle following PA. This is in line with the improved insulin sensitivity noted at the whole-body level after both ONE and MICRO [13].

Contrary to our hypothesis, we observed no functional adaptations in mitochondrial oxidative capacity in the presence of CHO-associated substrates following both PA interventions. This suggests that in the short-term, short-frequent bouts of PA can trigger changes in the regulation of gene expression in pathways associated with the regulation of metabolism, although these changes do not translate into higher mitochondrial function. This further suggests that mechanisms other than adaptations in mitochondrial oxidative capacity may contribute to the increase in 24 h CHO oxidation we observed previously in MICRO [13]. Because short-frequent bouts of brisk walking were completed throughout the day, participants were in the postprandial state. Therefore, the greater oxidative rates of CHO may be the result of CHO availability to provide energy for movement. Of note, substrate-driven LEAK respiratory state in the presence of CHO-associated substrate was negatively correlated with sleeping CHO oxidation (an index of fasting CHO oxidation) and no associations were observed for OXPHOS and ETS mitochondrial respiratory states. A potential early mechanism for adaptations in mitochondrial function may be the decrease in substrate-driven LEAK respiration through a more robust electron transport chain in which more protons are used for ATP production. Potential mitochondrial adaptations to MICRO may therefore take more time to develop. Longer-term studies will be needed to test if short-frequent bouts of PA can ultimately enhance overall skeletal muscle mitochondrial function.

We found that ONE, but not MICRO, resulted in a higher RCR when permeabilized muscle fibers were provided lipid-associated substrates, suggesting an improved coupling efficiency. The observed higher RCR in the presence of lipid-associated substrates may be due to a decrease in LEAK state FCR, which indicates a reduced contribution of the LEAK respiratory state to overall electron flux and improved coupling efficiency because fewer protons are lost to non-coupled respiration. Besides one study testing the effect of an acute 60 min bout of PA in 21 healthy untrained male participants [32], the evidence for acute and short-term adaptions in mitochondrial oxidative capacity are to our knowledge limited. Therefore, it is interesting to note that adaptations can be detected after 4 days of PA. In addition, this improved mitochondrial oxidative capacity for lipids, along with the molecular adaptations favoring substrate oxidation, may contribute to the higher 24 h FA oxidation we observed previously in response to a single-continuous bout intervention [13].

The daily single-continuous bout of intervention also involved more biological pathways that were enriched with DEGs than the short-frequent PA bouts intervention (Figure 3). These include the AMPK Signaling pathway, which is sensitive to changes in ATP turnover, the Calcium Signaling pathway, which is sensitive to oscillations in calcium concentrations from skeletal muscle contractile function, and the HIF1α Signaling pathway, which is sensitive to the intracellular partial pressure of oxygen [15]. Additionally, pathways associated with skeletal muscle nutrient uptake, growth, differentiation, survival, and the oxidative stress response were enriched after the daily single-continuous bout of PA intervention, including the Insulin Receptor Signaling, mTOR, ERK/MAPK Signaling, and NRF2-mediated Oxidative Stress Response pathways. This constellation of enriched pathways after the single-continuous PA bout intervention is known to be associated with PGC-1α [33,34,35,36], a master regulator of mitochondrial biogenesis [37], which may explain why we observed higher mitochondrial coupling after ONE. Of note, these results are confounded by an energy deficit, as energy expended during PA was not replaced and future studies will need to confirm if similar results are observed under stable energy balance.

The enrichment of several PA-associated transcriptional response pathways may be due to the sustained nature of the single-continuous PA bout intervention more than total active time. The longer bout length may create a greater challenge to skeletal muscle homeostasis, therefore eliciting a greater adaptive stress response. This might be particularly true for the Oxidative Phosphorylation pathway because, when comparing both PA interventions together, this pathway was predicted to be less expressed in the short-frequent bouts intervention. However, the Calcium Signaling pathway was predicted to be positively expressed following the exposure to frequent active breaks, suggesting that the frequency rather than the length of the PA bouts may impact this pathway. Sarcolemma depolarization during muscle contractions increases cytosolic Ca++ thereby activating the Calcium Signaling pathway. The frequent nature of the PA bouts may trigger waves of cytosolic Ca++ release that lead to increased expression of the Calcium Signaling pathway, which has been shown to be associated with mitochondrial biogenesis [38].

A strength of the study was the randomized cross-over design of the interventions; each participant served as their own control and all females were studied in the same phase of their menstrual cycle. We were able to compare the respective effects of interrupting sedentary time from increased energy expenditure and/or PA by including a time-matched control. Compliance with each intervention period was verified with PA monitors. The results of this ancillary study should also be considered along with several limitations. Muscle biopsies were harvested at two different time intervals after the last bout of PA (21 h post the single-continuous PA bout in ONE and 14 h after the last bout of the short-frequent PA bouts in MICRO). The current experiments quantified mitochondrial respiratory capacity and not the actual respiration directly following completion of each 4-day PA trial. Additionally, the sample size for RNA sequencing was small and the RNA sequencing results were not confirmed with RT-quantitative PCR due to limitations of muscle tissue availability.

## 5. Conclusions

Breaking up SB with short-frequent bouts of PA spread throughout the day over the short-term (4 days) is a sufficient stimulus to promote changes in the regulation of skeletal muscle gene expression in pathways associated with substrate oxidation. These changes were similar to those elicited by a time-matched daily single-continuous bout of PA, suggesting that total daily active time and/or energy expenditure is a primary trigger for those pathways. However, these molecular changes translated into an improved capacity for mitochondrial fat oxidation and were associated with higher expression in muscle contraction transcription signaling pathways only when PA was performed as a single-continuous bout. In contrast, the Calcium Signaling pathway was only activated when PA was completed as multiple short-frequent bouts. Therefore, this study showed for the first time that biological pathways in skeletal muscle are differentially regulated by the terms of PA, i.e., total active time, bout length, and bout frequency. Future studies will need to determine whether these differential responses are sustained over time and translate into differential functional and whole-body changes.

## Figures and Tables

**Figure 1 nutrients-14-00454-f001:**
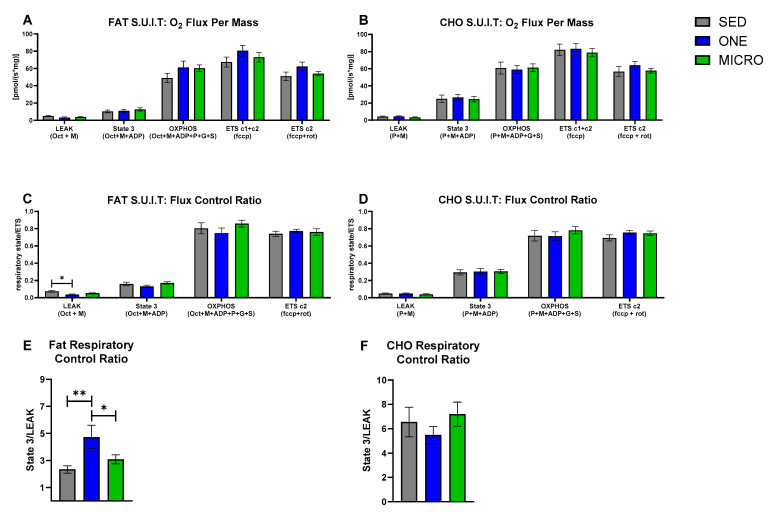
Carbohydrate- and fat-associated mitochondrial respiratory states, flux control, and respiratory control in permeabilized skeletal muscles fibers. Substrate–uncoupler–inhibitor titration protocols associated with carbohydrate (CHO) and fat mitochondrial substrates were used for the evaluation of mitochondrial respiratory states, flux control ratio (FCR), and respiratory control ratio (RCR) in permeabilized skeletal muscle fibers collected in the fasting state from adults with overweight or obesity (*n* = 19, 9 m/10 f) after completing three 4-day study interventions, SED: sedentary control, ONE: single-continuous bout of 45 min/day of moderate-intensity physical activity, MICRO: hourly 5 min bouts of moderate intensity physical activity performed for 9 consecutive hours per day. (**A**) Fat-associated mitochondrial respiratory states: LEAK: octanol (Oct) + malate (M); State 3: Oct + M + adenosine diphosphate (ADP); oxidative phosphorylation (OXPHOS): Oct + M + ADP + Pyruvate (P) + Glutamate (G) + succinate (S); Electron transport system (ETS) through complex 1 and 2 (ETS c1 + c2): fccp (carbonyl cyanide-p-trifluoromethoxyphenylhydrazone); ETS through complex 2 (ETS c2): fccp + rotenone (rot). (**B**) CHO-associated mitochondrial respiratory states: LEAK: P + M; State 3: P + M + ADP; OXPHOS: P + M + ADP + G + S; ETS c1 & c2: fccp; ETS c2: fccp + rot. The FCR was calculated for each respiratory state by normalizing the respiratory state to ETS c1 &c2 which quantifies the contribution of each respiratory state to overall electron flux, independent of mitochondrial content, mitochondrial preparations, and experiment-specific assay conditions. (**C**) Fat-associated FCR. (**D**) CHO-associated FCR. The RCR quantifies the degree of coupled respiration by State 3/LEAK. (**E**) Fat-associated RCR. (**F**) CHO-associated RCR. Data are mean ± SEM. Significant between-interventions * *p* < 0.05, ** *p* < 0.01.

**Figure 2 nutrients-14-00454-f002:**
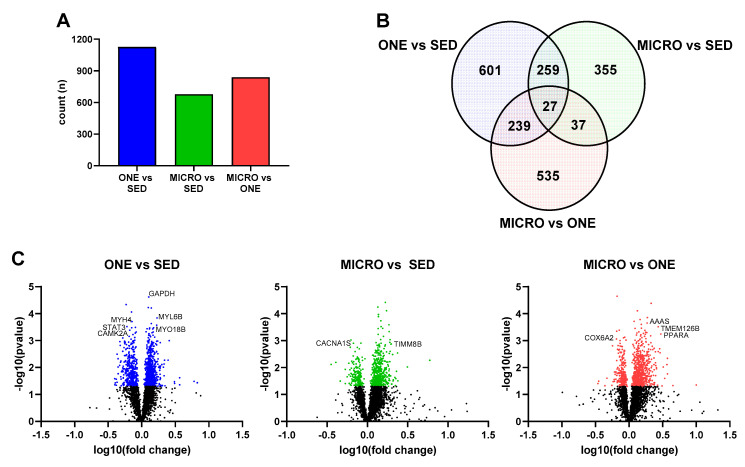
Comparison of muscle gene expression signatures in fasting skeletal muscle biopsies. (**A**) Counts of differentially expressed genes by comparison, (**B**) Venn diagrams displaying the counts of differentially expressed genes for each comparison, (**C**) Volcano plots of differentially expressed genes. Genes related to metabolic outcomes selected from top 30 indicated on plot. Skeletal muscle biopsies were collected in the fasting state from adults with overweight or obesity (*n* = 8, 4 m/4 f) after completing three 4-day study interventions: SED: sedentary control, ONE: single-continuous bout of 45 min/day of moderate-intensity physical activity, MICRO: hourly 5 min bouts of moderate intensity physical activity performed for 9 consecutive hours per day.

**Figure 3 nutrients-14-00454-f003:**
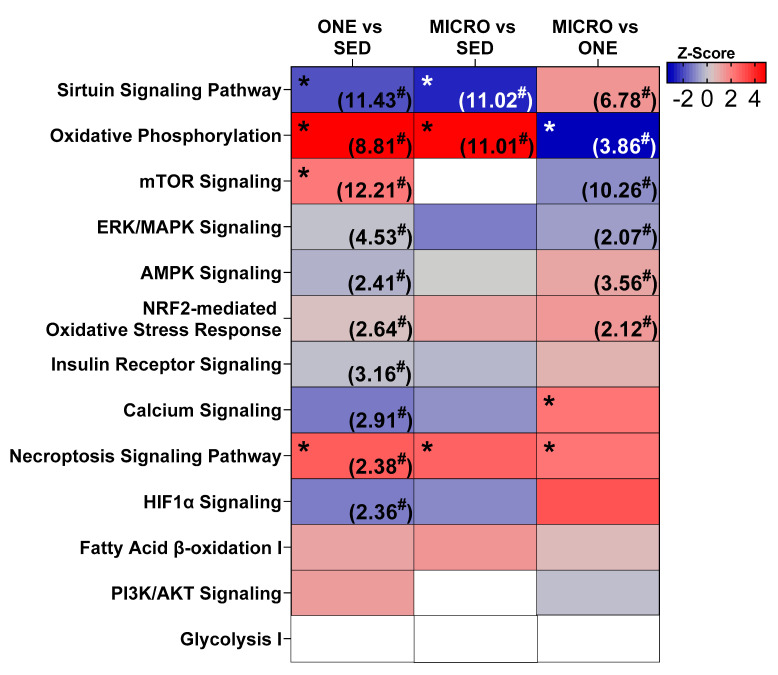
Comparison of metabolic and exercise-responsive pathways. Pathway enrichment was calculated as −log10(FDR) for each comparison (value indicated in parentheses). For pathway ranking, total FDR score was calculated as the sum across the row for each FDR score. Pathway activation state colored by z-score (white = N/A, red = activated, blue = inhibited). Some (*n* = 8, 4 m/4 f) of the skeletal muscle samples were analyzed by RNA sequencing to identify differentially expressed genes and pathway analysis was performed with Ingenuity Pathway Analysis (IPA, Qiagen Inc., Hilden, Germany). Skeletal muscle biopsies were collected in the fasting state from adults with overweight or obesity (*n* = 8, 4 m/4 f) after completing three 4-day study interventions: SED, sedentary control; ONE, single-continuous bout of 45 min/day of moderate-intensity physical activity; MICRO, hourly 5 min bouts of moderate intensity physical activity performed for 9 consecutive hours per day. * Significant z-score: *p* < 0.05. ^#^ Significant pathway enrichment: *p* < 0.05.

**Figure 4 nutrients-14-00454-f004:**
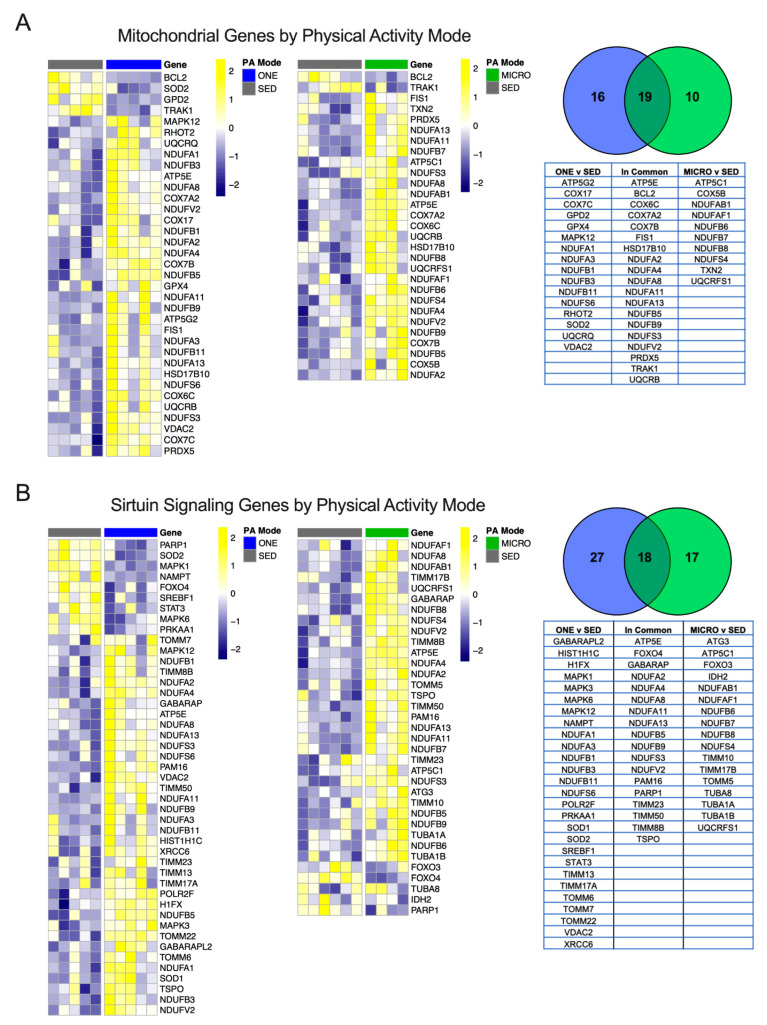
Mitochondrial and Sirtuin Signaling genes unique to physical activity mode. Significantly different genes between either ONE vs. SED or MICRO vs. SED belonging to the mitochondria (**A**) or the Sirtuin Signaling pathway (**B**) were plotted by heatmap. All genes presented are significant following multiple testing correction (FDR *p* ≤ 0.05). PA, physical activity; SED, sedentary control; ONE, single-continuous bout of 45 min/day of moderate-intensity physical activity; MICRO, hourly 5 min bouts of moderate intensity physical activity performed for 9 consecutive hours per day.

**Table 1 nutrients-14-00454-t001:** Participant characteristics.

	Muscle Mitochondrial Respiration (Oroboros)	Muscle Molecular Adaptations (RNA Seq)
N (#)	19 (9 m/10 f)	8 (4 m/4 f)
Age (years)	32.2 ± 6.4	34.3 ± 8.4
BMI (kg/m^2^)	30.3 ± 3.0	29.9 ± 2.4
Body Mass (kg)	85.2 ± 10.9	84.8 ± 11.6
FFM (kg)	55.2 ± 10.6	55.9 ± 11.0
FM (kg)	30.0 ± 7.1	28.9 ± 8.3
Fat mass (%)	35.3 ± 7.7	34.2 ± 8.8
Fasting Glucose (mg/dL)	87.8 ± 4.6	86.6 ± 4.8
Fasting Triglyceride (mg/dL)	102.7 ± 52.8	115.9 ± 67.5
HOMA-IR (SED day fasting)	1.4 ± 0.7	1.2 ± 0.6
IPAQ self-reported sitting time (h/day)	10.6 ± 3.3	8.5 ± 3.0

Data are presented as mean ± SD. N, number of participants; BMI, body mass index; kg, kilograms; FFM, fat-free mass; FM, fat mass; HOMA-IR, homeostatic model assessment of insulin resistance calculated from the blood draw collected on the morning of day 4 during the habitual sedentary behaviors (SED) intervention; IPAQ, international physical activity questionnaire.

**Table 2 nutrients-14-00454-t002:** Relationship between mitochondrial oxidative capacity and whole-body metabolic measures.

Dependent Variable	Independent Variable	Intervention	Slope Estimate	*p*-Value	*p* for Interaction
RCR fat	Fasting Insulin	SED	0.08	0.96	0.003 **
		ONE	0.65	0.008 ^##^	
		MICRO	0.24	0.02 ^#^	
		SED-ONE	−0.64	0.015 ^#^	
		SED-BREAK	−0.24	0.05 ^#^	
		ONE-BREAK	0.40	0.03 ^#^	
RCR fat	Insulin/Glucose	SED	4.27	0.58	0.005 **
		ONE	58.11	0.001 ^##^	
		MICRO	23.24	0.012 ^#^	
		SED-ONE	−53.84	0.002 ^##^	
		SED-BREAK	−18.97	0.07	
		ONE-BREAK	34.87	0.03 ^#^	
CHO LEAK	Sleeping CHO (g/min)	SED	−8.63	0.56	0.077 *
		ONE	27.24	0.17	
		MICRO	−36.06	0.03 ^#^	
		SED-ONE	−35.87	0.11	
		SED-BREAK	27.45	0.31	
		ONE-BREAK	63.32	0.03 ^#^	
CHO OXPHOS	Sleeping CHO (g/min)	SED	42.22	0.76	0.401
		ONE	−97.52	0.55	
		MICRO	−320.65	0.15	
		SED-ONE	139.74	0.53	
		SED-BREAK	362.86	0.18	
		ONE-BREAK	223.13	0.42	
CHO ETS c1 + c2	Sleeping CHO (g/min)	SED	−36.61	0.78	0.194
		ONE	229.91	0.15	
		MICRO	−200.32	0.33	
		SED-ONE	−266.53	0.18	
		SED-BREAK	163.71	0.49	
		ONE-BREAK	430.24	0.09	
FAT LEAK	Sleeping FAT (g/min)	SED	−2.47	0.94	0.919
		ONE	−8.59	0.81	
		MICRO	16.99	0.75	
		SED-ONE	6.11	0.89	
		SED-BREAK	−19.47	0.75	
		ONE-BREAK	−25.59	0.68	
FAT OXPHOS	Sleeping FAT (g/min)	SED	−105.42	0.71	0.357
		ONE	−460.21	0.15	
		MICRO	344.18	0.46	
		SED-ONE	354.79	0.41	
		SED-BREAK	−449.60	0.41	
		ONE-BREAK	−804.40	0.16	
FAT ETS c1 + c2	Sleeping FAT (g/min)	SED	−61.82	0.85	0.115
		ONE	−170.15	0.57	
		MICRO	936.09	0.04	
		SED-ONE	108.33	0.80	
		SED-BREAK	−997.91	0.08	
		ONE-BREAK	−1106.24	0.04	

Spearman correlational analysis between fasting insulin concentration and mitochondrial coupling in the presence of lipid-associated substrates stratified by intervention. A linear mixed-effect model was fit to the dependent variable for each trial simultaneously while accounting for the correlation of repeated observations from the same participant. Under the linear mixed-effect model, slopes of the dependent variable vs. independent variable were estimated and between-group comparison of the slopes were conducted. The between-group difference in slope assesses the difference between the two groups with respect to the correlation. SED, sedentary control; ONE, single-continuous bout of 45 min/day of moderate-intensity physical activity; MICRO, hourly 5 min bouts of moderate intensity physical activity performed for 9 consecutive hours per day; g/min, gram per minute substrate oxidation. Significant model interaction: * *p* ≤ 0.05, ** *p* ≤ 0.01. Significant between-intervention: ^#^
*p* ≤ 0.05, ^##^
*p* ≤ 0.01.

## Data Availability

Data will be made available by reasonable request through email.

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
