# Peer review of "Short-Term Adaptations in Skeletal Muscle Mitochondrial Oxidative Capacity and Metabolic Pathways to Breaking up Sedentary Behaviors in Overweight or Obese Adults"

_nutrients, 2022, doi:10.3390/nu14030454_

Round 1

Reviewer 1 Report

The article entitled “Short-Term Adaptations in Skeletal Muscle Mitochondrial Oxidative Capacity and Metabolic Pathways to Breaking Up Sedentary Behaviors in Adults with Overweight or Obesity” presents a human study comparing the influence of physical activity performed for 4 days by the subjects on the molecular adaptations in skeletal muscles. Physical activity was performed in two manners: short-term (up to 5 min every hour over a 9h period) and long-term (up to 45 min once during the day). The results show little differences between the two and the authors point out that to better assess the issue longer-term study should be conducted.

Some suggestions are listed below:

  1. I suggest correcting the title to: Short-Term Adaptations in Skeletal Muscle Mitochondrial Oxidative Capacity and Metabolic Pathways to Breaking Up Sedentary Behaviors in Overweight or Obese Adults
  2. English requires corrections throughout the text and vocabulary should be simplified where possible to make it easier to understand by the reader.
  3. Please correct some minor oversights e.g. abbreviations should be explained at first use (CHO), “hr” is used in some places, and in some it is just “h” for an hour, the lower casing of numbers/letters is missing
  4. The introduction should be corrected. It can be simplified for the better understanding by the reader (avoid overcomplication of sentences) and information should be put in better logical order so the reader may easily follow the authors' thoughts.
    g.
    line 43: replace “High amounts of time spent in” with „prolonged”
    line 45: replace „Total 45 daily sedentary time as well as prolonged sedentary activities have been” with „It has been”
    line 51: „observed as”
    lines 55-59: Please move it to line 52 before the last sentence of the paragraph.
    line 60: „short PA bouts” please specify its time. is it again up to 5 min?
    line 64: please give examples of genes that are differentially expressed
    line 65: Could you provide some more findings supporting the thesis about the promotion of molecular adaptations in skeletal muscle? the paragraph would benefit from elaborating this issue
    lines 66-69: please move the sentence to line 54 as the last sentence of the first paragraph
  5. line 202: was any specific kit used for isolation? please give the name, not just a manufacturer.
  6. Discussion would benefit from better structuring and simplifying the sentences.
  7. The conclusions are oversimplified and lack a “take-home message”.
  8. The article would benefit from re-writing to clarify the thesis, simplify the text to make it easier to follow by the reader, and point out which conclusions concern this study and which concern previous ones.

Author Response

Reviewer #1

  1. I suggest correcting the title to: Short-Term Adaptations in Skeletal Muscle Mitochondrial Oxidative Capacity and Metabolic Pathways to Breaking Up Sedentary Behaviors in Overweight or Obese Adults
    1. The title has been changed.
  2. English requires corrections throughout the text and vocabulary should be simplified where possible to make it easier to understand by the reader.
    1. We appreciate the reviewer’s comments. The entire manuscript has been revised. We hope these changes will help the reader to better follow the narrative.
  3. Please correct some minor oversights e.g. abbreviations should be explained at first use (CHO), “hr” is used in some places, and in some it is just “h” for an hour, the lower casing of numbers/letters is missing
    1. The abbreviation was added to line 64 at the first use.
    2. Throughout the document “hr” is the abbreviation used.
  4. The introduction should be corrected. It can be simplified for the better understanding by the reader (avoid overcomplication of sentences) and information should be put in better logical order so the reader may easily follow the authors' thoughts.
    1. line 43: replace “High amounts of time spent in” with „prolonged” and line 45: replace „Total 45 daily sedentary time as well as prolonged sedentary activities have been” with „It has been”
      1. Thank you for the suggestion. These have been changed.
    2. line 51: „observed as”
      1. The introduction was revised, and this sentence was removed.
    3. lines 55-59: Please move it to line 52 before the last sentence of the paragraph.
      1. This revision was completed. (ln 52)
    4. line 60: „short PA bouts” please specify its time. is it again up to 5 min?
      1. This revision was completed. The bout length for this study was 2-min (ln 77)
    5. line 64: please give examples of genes that are differentially expressed
      1. ln 72-76, we outlined a few of the differentially expressed genes that were activated after acute and short-term trials of breaking up sedentary behaviors with short-frequent bouts of physical activity.
    6. line 65: Could you provide some more findings supporting the thesis about the promotion of molecular adaptations in skeletal muscle? the paragraph would benefit from elaborating this issue
      1. Ln 68-82 we emphasize the previously published evidence on the molecular adaptations observed in response to acute and short-term interventions of breaking up SB with short-frequent bouts of physical and the gap in knowledge that no study has compared the transcriptional response to frequent active bouts to break up prolonged sedentary periods to a time-matched single continuous bout of physical activity that this paper discusses.
    7. lines 66-69: please move the sentence to line 54 as the last sentence of the first paragraph
      1. This revision was completed. (ln 66)
    8. line 202: was any specific kit used for isolation? Please give the name, not just a manufacturer.
      1. This revision was completed. (ln 206)
  1. Discussion would benefit from better structuring and simplifying the sentences.
    1. The entire discussion was edited, and sentences simplified to help reader comprehension.
  2. The conclusions are oversimplified and lack a “take-home message”.
    1. The take home message of the manuscript is that biological pathways in skeletal muscle are differentially regulated by the terms of physical activity (bout length and frequency). This message was added to the conclusion. (ln 458-462).

Reviewer 2 Report

The manuscript is an interesting original research investigating the effects of a 4-day of two different walking bouts – 5 min each hour up to 9 h vs. 45 min continuous per day – in metabolic responses in overweight or obese adult men and women. This work contributes to the field of study reporting a greater response in mitochondrial coupling in the presence of lipid-associated substrates and in the enrichment of signaling pathways such as Ca++, mTOR, AMPK or HIFa after the continuous, comparing with short, brisk walking. Also, the current study presents similar results for both physical activity bouts in terms of respiratory rates, oxidative phosphorylation and sirtuin signaling. The main limitation of the study is not having confirmed protein expression by Western blot.

Minor improvements are needed for publication:

  1. Was the data check for normal distribution? The authors should include the information about the test in the statistical analysis section.
  2. The authors should use the same format for p values, since in the abstract appear without 0 before the point and in the text and tables with it.
  3. The authors should check if the full name of all abbreviations has been indicated in the first mention in the manuscript (e.g. CHO – Line 64 – is defined in Line 77).
  4. The authors should correct some misspellings like PGC1a (Lines 426-427) (vs. PGC-1a).

Author Response

Reviewer #2

  1. Was the data check for normal distribution? The authors should include the information about the test in the statistical analysis section.
    1. This has been added. (ln 230)
  2. The authors should use the same format for p values, since in the abstract appear without 0 before the point and in the text and tables with it.
    1. The revision was completed.
  3. The authors should check if the full name of all abbreviations has been indicated in the first mention in the manuscript (e.g. CHO – Line 64 – is defined in Line 77).
  1. The abbreviation was added to line 64 at the first use.
  1. The authors should correct some misspellings like PGC1a (Lines 426-427) (vs. PGC-1a).
    1. The revision was completed. (ln 417).
